# An elasticity-curvature illusion decouples cutaneous and proprioceptive cues in active exploration of soft objects

**Chang Xu**, **Yuxiang Wang**, **Gregory J. Gerling**\*

School of Engineering and Applied Science, University of Virginia, Charlottesville, Virginia, United States of America

\* gg7h@virginia.edu

**Data Availability Statement:** All relevant data can be found at https://doi.org/10.6084/m9.figshare.14161313.

**Funding:** This work is supported in part by grants from the National Science Foundation (IIS-

## Abstract

Our sense of touch helps us encounter the richness of our natural world. Across a myriad of contexts and repetitions, we have learned to deploy certain exploratory movements in order to elicit perceptual cues that are salient and efficient. The task of identifying optimal exploration strategies and somatosensory cues that underlie our softness perception remains relevant and incomplete. Leveraging psychophysical evaluations combined with computational finite element modeling of skin contact mechanics, we investigate an illusion phenomenon in exploring softness; where small-compliant and large-stiff spheres are indiscriminable. By modulating contact interactions at the finger pad, we find this elasticity-curvature illusion is observable in passive touch, when the finger is constrained to be stationary and only cutaneous responses from mechanosensitive afferents are perceptible. However, these spheres become readily discriminable when explored volitionally with musculoskeletal proprioception available. We subsequently exploit this phenomenon to dissociate relative contributions from cutaneous and proprioceptive signals in encoding our percept of material softness. Our findings shed light on how we volitionally explore soft objects, i.e., by controlling surface contact force to optimally elicit and integrate proprioceptive inputs amidst indiscriminable cutaneous contact cues. Moreover, in passive touch, e.g., for touch-enabled displays grounded to the finger, we find those spheres are discriminable when rates of change in cutaneous contact are varied between the stimuli, to supplant proprioceptive feedback.

## Author summary

How do we differentiate soft objects by touch, as we do in judging the ripeness of fruit? Our understanding of how material softness is perceptually encoded remains incomplete. This study investigates an illusion phenomenon that occurs in discriminating material compliances. We find that small-compliant and large-stiff spheres are naturally indistinguishable when pressed into a stationary finger, but readily discriminable when pressed upon. This phenomenon illuminates an interplay within our somatosensory system, in particular, between cutaneous responses from skin receptors and proprioceptive feedback traditionally tied to joint movements. It also reveals how our movements optimally evoke

1908115) and National Institutes of Health (NINDS R01NS105241) to GJG. The funders had no role in study design, data collection and analysis, decision to publish, or preparation of the manuscript.

**Competing interests:** The authors have declared that no competing interests exist.

these cues to inform our percept of softness. Understanding how softness is encoded at skin contact is key to designing touch-enabled displays. Moreover, our approach is to computationally evaluate combinations of stimulus elasticity and curvature in modeling space prior to empirical experiments with human subjects.

## Introduction

We integrate a multimodal array of sensorimotor inputs in the everyday perception of our natural environment. Along with vision and audition, our sense of touch is essential in interactions involving dexterous manipulation, affective connections, and naturalistic exploration [1–4]. For example, we routinely judge the ripeness of fruit at the grocery store, caress the arm of a spouse to offer comfort, and stroke textiles to gauge their roughness and softness [5–7]. We seamlessly do so by recruiting sensorimotor inputs, fine-tuning motor control strategies, comparing current percepts to our prior expectations, and updating internal representations [8].

Historically, tactile illusions have revealed inherent interdependencies of our sensorimotor and perceptual systems. Among the many illusions identified [9,10], the "size-weight" illusion is particularly well-known. It involves picking up two objects of identical mass but of varied volume, and indicates that the smaller object is generally perceived as heavier [11]. The size-weight illusion reveals a separation of our sensorimotor and perceptual systems in estimating an object's mass. In particular, while our sensorimotor system adapts to the mismatch between the predicted and actual signals to dynamically adjust our exploratory motions, our perceptual system recalibrates the size-weight relationship more gradually on a different time scale [9,12,13]. Another intriguing illusion regards our perception of curvature where a physically flat surface is manually explored along a lateral direction. Depending on the relative inward/outward motions of the surface and the observer's finger, the flat surface can be perceived as being convex or concave [9,14]. The curvature illusion reveals a poor spatial constancy of our somatosensory system, driven by a dissociation between cutaneous and proprioceptive inputs [4]. A further illusion, by analogy with the Aubert-Fleischl phenomenon in vision, indicates a possible misperception in speed by touch [15]. In particular, observers are asked to estimate the speed of a moving belt stimulus. Compared with tracking the stimulus with a guided arm movement, where the finger is moving along with the belt's motion (i.e., proprioception is available), observers can overestimate the stimulus speed by touching the stimulus with a stationary hand (i.e., tactile cues only). These and other illusions shed light upon interdependencies of our sensorimotor and perceptual systems, i.e., processing mechanisms for the perception of object properties, e.g., size, orientation, and movement, are distinct from those underlying the mediation of those properties in sensorimotor control [16–19]. Furthermore, tactile illusions can serve as a tool in engineering applications where human perception could be manipulated, e.g., the "size-weight" illusion could be exploited to create particular stimuli in virtual reality whose physical properties may be perceived as changing during interactions. Meanwhile, illusions have also been considered as a metric to evaluate virtual environments by correlating the perceived realism with the illusion strength [9].

Among the many dimensions of touch, which include surface roughness, stickiness, geometry, and others, our perception of softness is central to everyday life [2]. Our understanding of tactile compliance, a key dimension of an object's "softness," remains incomplete. This percept is informed by some combination of cutaneous inputs from mechanosensitive afferents signaling skin deformation and proprioceptive inputs signaling body movements. Efforts to define the precise cues within skin deformation and body movements have focused on contact area at

the finger pad [20–24], spatiotemporal deformation of the skin's surface [25–27], and kinesthetic inputs of displacement, force, and joint angle [28–31]. Such an array of sensory contact inputs, mediated by independent cortical mechanisms, are recruited and integrated in the primary somatosensory cortex, and form the perceptual basis from which compliances are recognized and discriminated [32]. That being said, it yet remains unclear which exploratory movements could elicit those perceptual cues that most optimally encode material softness.

Here, we investigate a tactile illusion associated with softness perception, specifically, in exploring spherical stimuli with covaried elasticity and curvature. These physical attributes are routinely encountered, such as in judging the ripeness of spherical fruit. The illusion phenomenon is observed only in passive touch, when the finger is stationary and only non-distinct cutaneous cues of interior stress and gross contact areas are available for perception. The spheres, however, become readily discriminable when explored volitionally in active touch where finger proprioception is involved. The spheres therefore naturally dissociate relative contributions from cutaneous and proprioceptive cues in encoding softness, and shed light into how we volitionally explore compliant objects in everyday life.

## Results

We introduce a novel elasticity-curvature illusion where small-compliant and large-stiff spheres are perceived as indiscriminable in passive touch. These spheres are explored using single, bare finger touch. Our methodological paradigm is unique in that computational models of the skin's mechanics define the stimulus attributes prior to evaluation in human-subjects experiments. In particular, finite element models of the distal finger pad are used to develop elasticity-curvature combinations that afford non-differentiable cutaneous cues. Then, investigation of the mechanisms that underlie this potential illusory experience is done empirically with human-subjects via measurements of biomechanical interactions and evaluations of psychophysical responses. The results suggest that we use a force-controlled movement strategy to optimally evoke cutaneous and proprioceptive cues in discriminating softness.

First, the skin mechanics of the index finger are modeled with finite elements in simulated interactions with spherical stimuli. The models predict that small-compliant (10 kPa–4 mm) and large-stiff (90 kPa–8 mm) spheres will generate nearly identical cutaneous contact cues, which may render them indiscriminable in passive touch. In contrast, when the models simulate conditions of active touch, the resultant fingertip displacements with controlled force loads are found to be distinct, which may render them discriminable.

Next, driven by the model predictions, a series of biomechanical and psychophysical evaluations are conducted with human participants. The results reveal that these spheres are indeed indiscriminable when explored in passive touch with only cutaneous cues available. However, this phenomenon vanishes when cues akin to proprioception are systemically augmented by a participant's use of a force control movement strategy.

### Experiment 1: Computational modeling of the elasticity-curvature illusion

Finite element analysis was performed to simulate the skin mechanics of the bare finger interacting with compliant stimuli. The material properties of the model were first fitted to known experimental data. Then, numerical simulations were conducted with spherical stimuli of covaried radius (4, 6, and 8 mm) and elasticity (10, 50, and 90 kPa). In two interaction cases, the fingertip was moved and constrained to simulate active and passive touch, respectively. To help quantify the discriminability of the spheres, response variables were derived from the stress distributions at the epidermal-dermal interface, where Merkel cell end-organs of slowly adapting type I afferents and Meissner corpuscles of rapidly adapting afferents reside, as well as the

required fingertip displacements to a designated touch force. The former was deemed as the cutaneous cue [27,33–35]. The latter was associated with the proprioceptive cue where displacement approximates the change in muscle length and force tied to muscle tension [30,35–38].

In simulation of passive touch where only cutaneous cues are available, the compliant spheres deformed the surface of the skin distinctly for each combination of elasticity and radius (Fig 1). Spatial distributions of stress for both the finger pad and spheres were simulated to a steady-state load of 2 N. For all the nine spheres simulated, either an increase of the spherical radius or a decrease of the elasticity decreases the concentration of stress quantities at contact locations, with the lowest stress concentration for the 10 kPa-8 mm sphere and the highest for the 90 kPa-4 mm sphere (detailed in S1 Fig). Note that the 10 kPa-8 mm sphere was taken as the comparison case in the following analyses.

However, for certain elasticity-radius combinations, changes in the spheres' radii counteracted the changes in their elasticity, resulting in nearly identical stress distributions for cutaneous contact. Although the deformation of the stimuli differed vastly between the 10 kPa-4 mm (Fig 1A) and 90 kPa-8 mm spheres (Fig 1C), the surface deformation and stress distributions of the finger pad were quite similar. Specifically, stress distributions at the epidermal-dermal interface were nearly identical between the small-compliant (10 kPa-4 mm) and large-stiff (90 kPa-8 mm) spheres across all levels of load (Fig 2A). A similar case was demonstrated for the 10 kPa-4 mm and 90 kPa-6 mm spheres where the stress curves fairly well overlapped (Fig 2B), as compared to the distinct stimulus (Fig 2C).

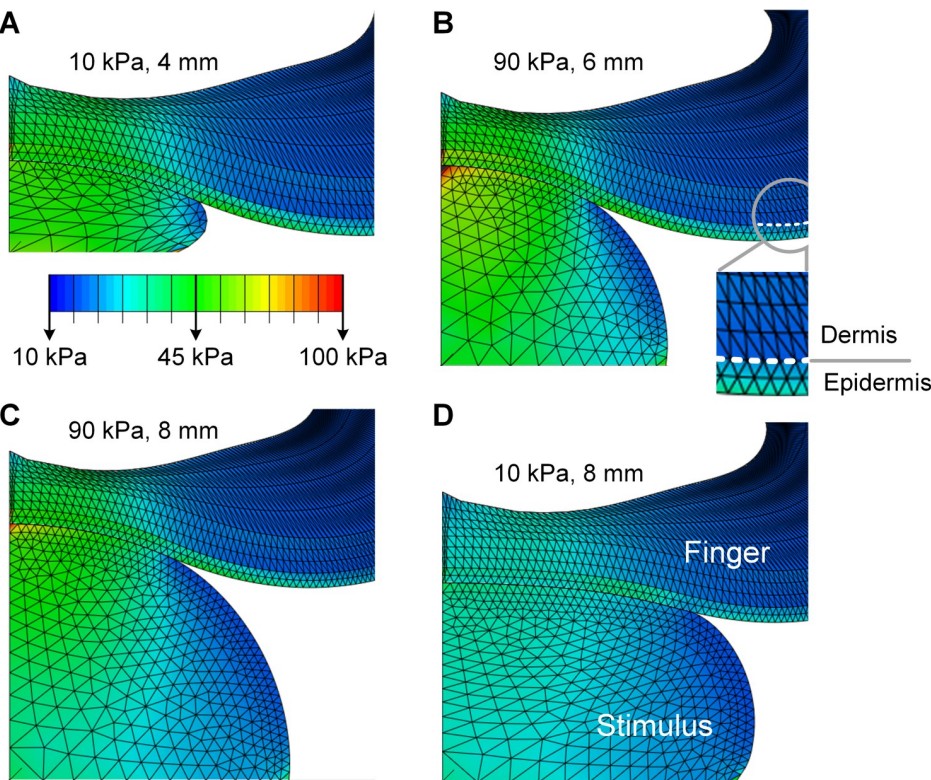

**Fig 1. Computational modeling of contact mechanics with compliant spheres.** Spatial distributions of stress are simulated at a load of 2 N for contact with spheres of (A)10 kPa-4 mm, (B) 90 kPa-6 mm, (C) 90 kPa-8 mm, and (D) 10 kPa-8 mm respectively. The epidermal-dermal interface was indicated in (B) and was consistently modeled for all simulation conditions. Although the deformation of the spherical stimuli differs greatly from (A) to (C), the resultant stress distributions and surface deflection at the finger pad are nearly identical.

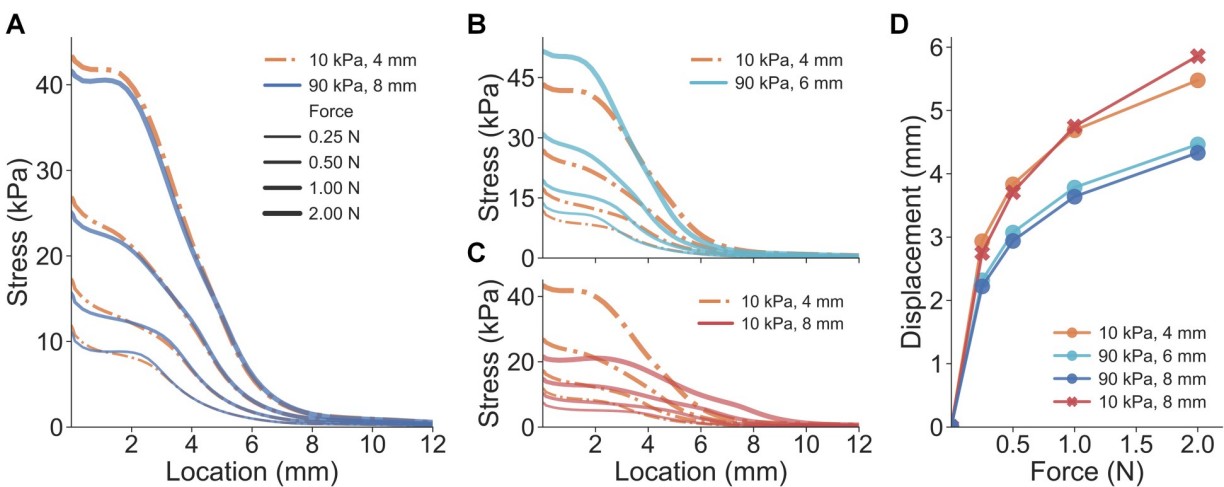

**Fig 2. Results of experiment 1: cues of cutaneous contact and proprioception.** (A) For the small-compliant (10 kPa-4 mm) and large-stiff (90 kPa-8 mm) spheres, stress distributions at the epidermal-dermal interface are nearly identical across all force loads. (B) Curves of stress distributions fairly well overlap for the 10 kPa-4 mm and 90 kPa-6 mm spheres. (C) Distinct stress distributions were obtained for spheres with the same elasticity but varied radii. (D) Proprioceptive cues of finger displacement are simulated for all force loads.

In addition to spatial distributions of stress, other response variables were also evaluated. The strain energy density (SED) at the epidermal-dermal interface and the deflection of the skin's surface were calculated and analyzed. Besides the stress/strain distributions, deflection of the skin surface–quantified by displacements at the node of the epidermis surface—is often considered as a cutaneous cue informing the change of contact area [20,22]. Similar to the results in Fig 2, SED distributions and skin surface deflection from the three spheres (10 kPa-4 mm, 90 kPa-6 mm, and 90 kPa-8 mm) were nearly inseparable, which were predicted to generate indiscriminable contact area cues upon contact (Fig 3, detailed in S1 and S2 Figs). In addition, mean values of cutaneous responses over the contact region were also similar between the three spheres (S3 Fig). These results demonstrate that small-compliant and large-stiff stimuli can generate nearly identical cutaneous contact cues, therefore, non-informative for discriminating compliances whereas proprioceptive cues may be useful. It indicates that in passive touch where only cutaneous cues are perceptible, one might be unable to differentiate the aforementioned spheres. Therefore, these three stimuli (10 kPa-4 mm, 90 kPa-6 mm, and 90 kPa-8 mm) were denoted as the "illusion case spheres."

In simulation of active touch, where both cutaneous and proprioceptive cues are available, an increase in either the radius or elasticity decreases the fingertip displacement given the same load (S2 Fig). Specifically, the force-displacement curve of the 10 kPa-4 mm sphere was clearly separable from the 90 kPa-8 mm sphere (Fig 2D). Additionally, spheres of the same elasticity yielded overlapping force-displacement curves, as opposed to spheres of different elasticity. These results demonstrate that distinct proprioceptive cues tied to fingertip displacement differ given the indentation of the small-compliant compared to the large-stiff spheres. In active touch, where cues tied to fingertip displacement are utilized, one might be able to perceptually discriminate those illusion case spheres (10 kPa-4 mm, 90 kPa-6 mm, and 90 kPa-8 mm) amidst non-differentiable cutaneous contact cues.

Besides analyzing response variables only at the steady-state, the stimulus-ramp phase was further simulated to evaluate how contact mechanics would derive responses during the dynamic contact (detailed in S1 Text). Overall, the illusion case spheres could still afford nearly identical cutaneous responses during the stimulus ramp (S3 and S7 Figs). The rate of change

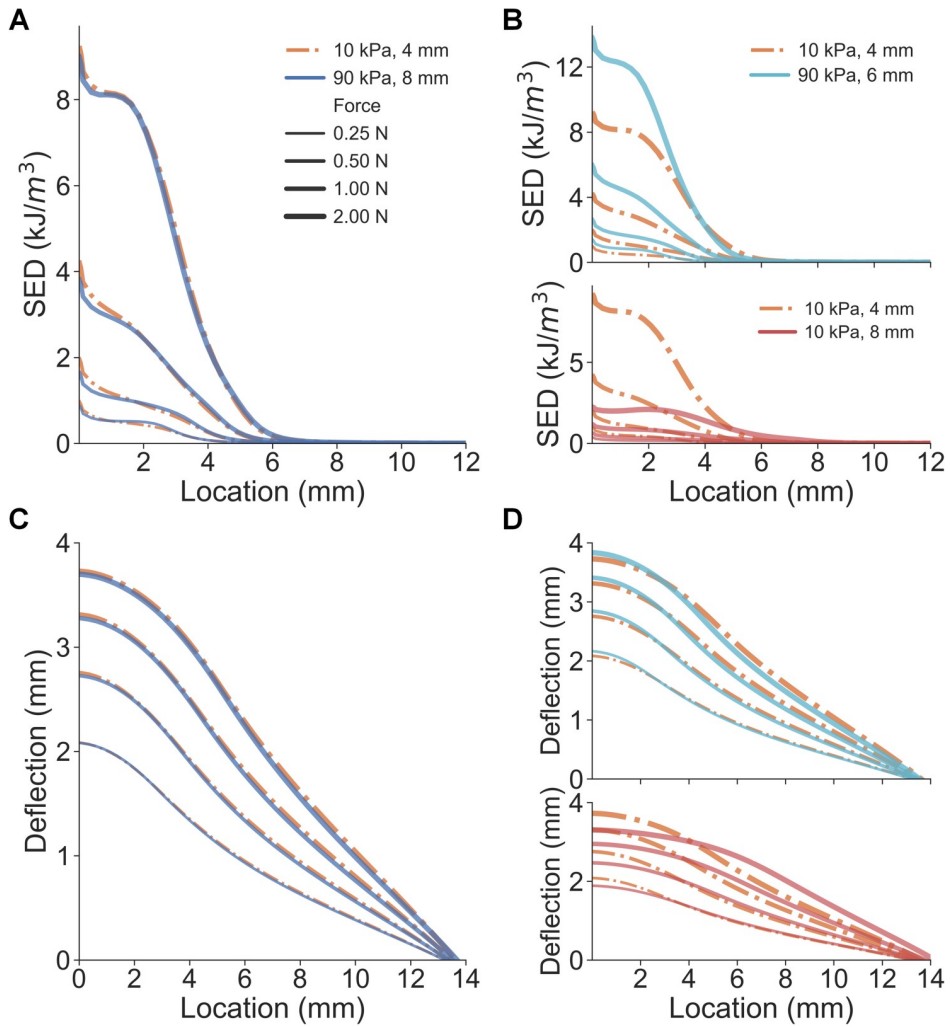

**Fig 3. Comparison of cutaneous cues between illusion and distinct spheres.** (A) Spatial distributions of SED are nearly identical for the small-compliant and large stiff spheres. (B) As opposed to the 10 kPa-8 mm sphere, SED distributions fairly well overlap between the 10 kPa-4 mm and 90 kPa-6 mm spheres. (C) Non-distinct surface deflection cues are obtained from the small-compliant and large stiff spheres. (D) Consistent with SED distributions, surface deflections overlap for the 10 kPa-4 mm and 90 kPa-6 mm spheres.

in stress distributions, SED, and surface deflection cues consistently overlap (S8 Fig). This indicates that, throughout contact time-course done *in silico*, similar afferent responses from both slowly and rapidly adapting mechanoreceptors might be elicited among the illusion case spheres, and thus, may render an illusory experience in discriminating their compliances.

## Experiment 2: Biomechanical measurement of cutaneous contact

Derived from the computational analysis in Experiment 1, we hypothesized that similar cutaneous contact cues might be observed among the illusion case spheres. To validate this prediction, we conducted biomechanical measurement experiments with human-subjects.

In particular, through a series of biomechanical measurements, the contact area between the finger pad and stimulus was quantified to determine if the illusion case spheres would generate similar cutaneous contact profiles. Contact area was measured directly, using an ink-based procedure [26]. Measured contact area is commensurate with the cutaneous cues

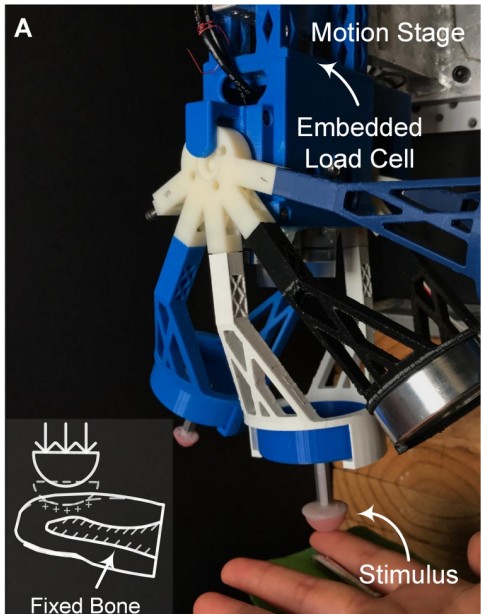
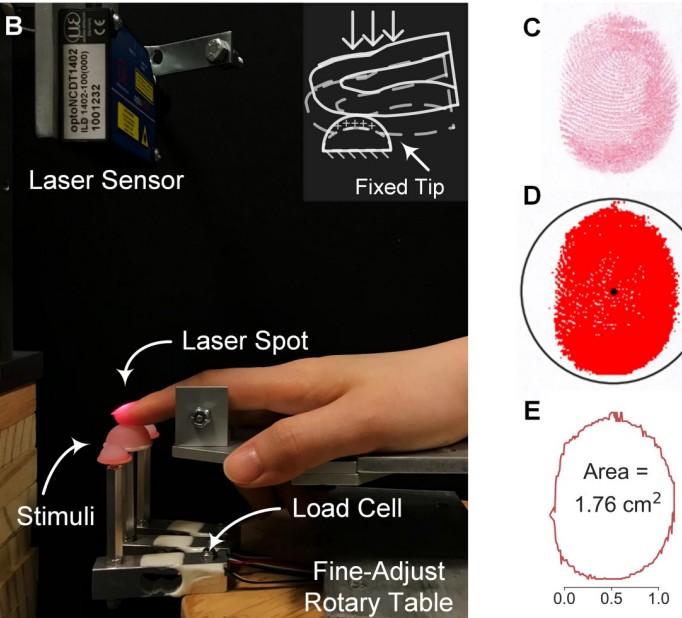

**Fig 4. Experimental setup and ink-based contact area analysis.** (A) For passive touch, the compliant stimulus is indented into the fixed finger pad by the motion stage. Contact force is measured by the embedded load cell. (B) For active touch, the designated stimulus is fixed and volitionally contacted by the index finger. Touch force is measured by the load cell underneath and fingertip displacement is captured by the laser sensor. (C) Contacted fingerprints are stamped and digitized for analysis. (D) The contact region is identified and color-thresholded. (E) Contact area is calculated based on the exterior outline and scaled pixels.

predicted in the finite element simulation. In the simulation, stress/strain distributions at contact locations and the skin surface deflection are quantified as cutaneous cues. In the experiments, contact area is derived from a contiguous area on the skin surface with a super-threshold contact pressure [22,23]. Furthermore, the deflection of the skin surface is the contour of a deflection profile in the contact plane [27,35].

In passive touch, where compliant stimuli are indented into a fixed fingertip, a customized indenter was utilized (Fig 4A). Participants ($n = 10$) were instructed to rest their forearm and wrist on a stationary armrest and the index finger was constrained. Each of the four spheres (three illusion case stimuli and one distinct stimulus) was indented into the finger pad with a triangle-wave force profile peaking at the desired level (1, 2, and 3 N). To quantify the contact area at the peak magnitude of indentation, an ink-based procedure was employed. The stamped finger pad was digitized (Fig 4C) and the contact region was color-enhanced (Fig 4D). The contact areas were then calculated based on the exterior outlines with scaled pixels (Fig 4E).

Non-distinct relationships of touch force and contact area are indeed observed in passive touch between the illusion case spheres across loading levels (Fig 5). By inspecting results from the example participant (Fig 5A), illusion case spheres (10 kPa-4 mm, 90 kPa-6 mm, and 90 kPa-8 mm) generated similar contact areas while the distinct sphere (10 kPa-8 mm) afforded higher contact areas. There was a significant difference between contact areas of the illusion case and distinct spheres across all force levels ($U = 0.0$, $p < 0.0001$, $d = 4.81$). In particular, data points for the three illusion cases were well clustered across all force levels (mean contact area: $0.90 \pm 0.12$ cm$^2$, mean $\pm$ SD), while the others were significantly distinct from them (mean contact area: $1.68 \pm 0.18$ cm$^2$). For all participants aggregated (Fig 5C), the force-contact area relation appeared to be consistent within an individual. Traces for the three illusion cases well overlapped (no significant difference detected) across all force levels, while the trace

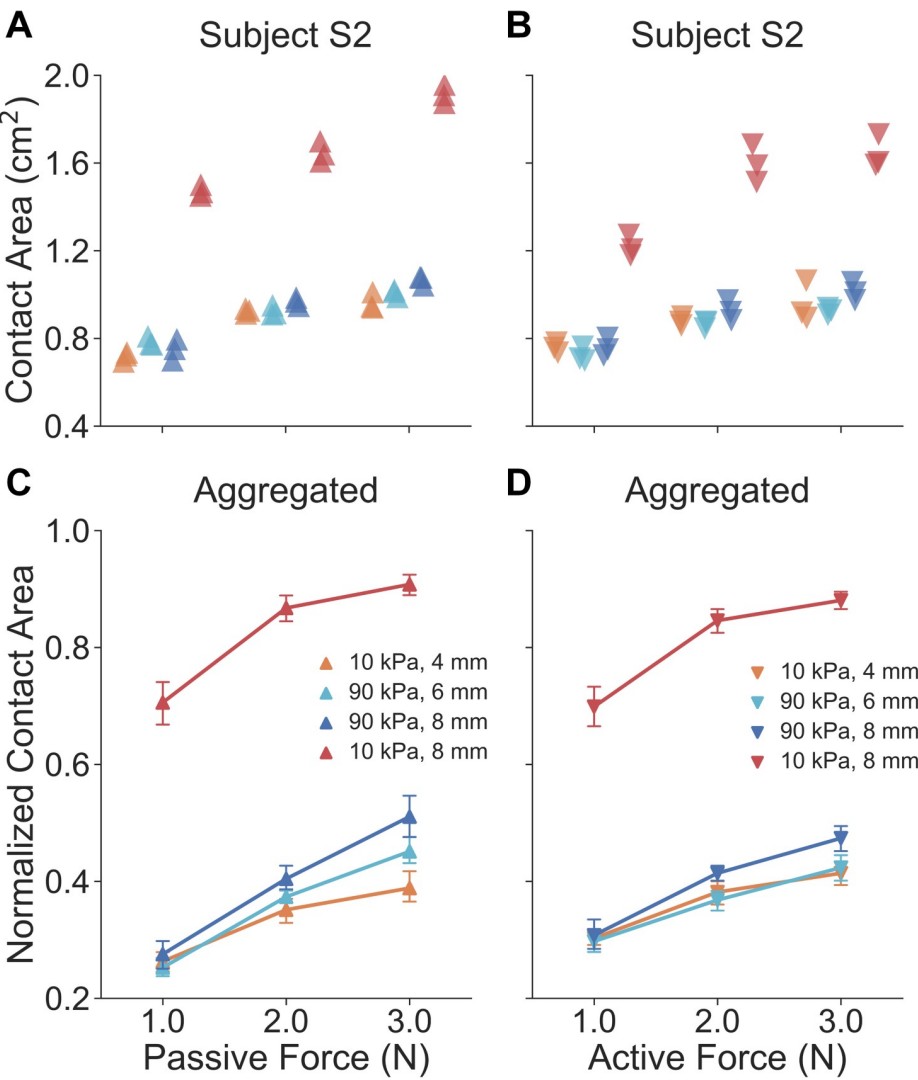

**Fig 5. Results of experiment 2: biomechanical measurements of contact area.** For a representative participant, in both (A) passive and (B) active touch, gross contact areas for illusion case spheres across all force levels are nearly identical, as opposed to the 10 kPa-8 mm sphere. Note that each data point represents the contact area measured from each indentation. For all participants aggregated, both in (C) passive and (D) active touch, curves of the illusion cases well overlap across all force levels, as opposed to the 10 kPa-8 mm sphere. Error bars denote 95% confidence intervals.

for the 10 kPa-8 mm sphere was distinct. Specifically, there was a significant difference between contact areas of the illusion case and distinct spheres across all force levels ($U = 87.0$, $p < 0.0001$, $d = 4.49$).

In active touch, where the finger volitionally touches the fixed compliant stimulus, an experimental setup was built as illustrated in Fig 4B. Participants ($n = 10$) were instructed to press their index finger down into a spherical stimulus without external constraint. A sound alarm was triggered to end each exploration when the touch force reached the desired level. After each exploration, the ink-based procedure was conducted to measure the contact area between the finger pad and stimulus.

Similar force-contact area relations were found in active touch as found in passive touch. Within a participant (Fig 5B), and similar to the passive touch experiments, the illusion case

spheres generated similar gross contact areas while the 10 kPa-8 mm sphere exhibited higher values. There was a significant difference between results of the illusion case and distinct spheres across all force levels ($U = 0.0$, $p < 0.0001$, $d = 3.73$). Specifically, the mean contact area for the three illusion cases is $0.87 \pm 0.10$ cm$^2$ while the other distinct stimulus derived a mean contact area of $1.48 \pm 0.20$ cm$^2$ across all force levels. For all participants aggregated (Fig 5D), traces for the illusion cases well overlapped (no significant difference detected), and the 10 kPa-8 mm sphere yields a much more distinct relationship. Specifically, there was a significant difference between the contact areas of the illusion case and distinct spheres across all force levels ($U = 0.0$, $p < 0.0001$, $d = 4.94$). Since cues tied to contact area are not significantly different, proprioceptive inputs evoked in active touch may be vital to discriminating the illusion case spheres.

### Experiment 3: Psychophysical evaluation of the elasticity-curvature illusion

The results of Experiment 2 support the hypothesis that cutaneous contact cues are not significantly different among illusion case spheres, for both passive and active touch. To evaluate whether there is a perceptual illusion in exploring these compliant spheres, we conducted psychophysical experiments with human-subjects.

Participants ($n = 10$) were first instructed to discriminate the illusion case spheres in passive touch. To further investigate the utility of temporal cues in augmenting our discrimination performance, the indentation force-rate was systematically modulated in three different experimental conditions. In the "passive same force-rate" task, where indentation rate was controlled at 1 N/s (Fig 6A), participants were not able to discriminate the stimuli (percentage of correct responses: 46.1% ± 5.7). In addition, the sensitivity measure $d'$ was also calculated under the assumption of differencing rule [39]. The mean $d'$ of 0.42 indicated a chance performance across all stimulus pairs (detailed in S4 Table). These illustrate that when only cutaneous cues are available, but their contact areas do not differ, these spheres indeed are indiscriminable.

Then, to evaluate the discriminability of these stimuli when adding proprioception to cutaneous contact, controlled force inputs were induced in passive touch in two separate cases. In the "passive inverse force-rate" task (Fig 6A), where the softer stimulus was indented "inversely" at a higher force-rate (2 N/s) than the harder stimulus (0.5 N/s), participants were still unable to discriminate the compliances with a percentage of correct responses of 52.8% ± 6.7. However, this result (with all participants aggregated) was significantly higher compared with the "passive same force-rate" condition ($U = 24.0$, $p < 0.05$, $d = 1.03$). The mean $d'$ value of 1.19 across stimulus pairs also indicated an improved, but still poor discrimination sensitivity under this condition (detailed in S4 Table). This aligns with prior work demonstrating that participants exhibit a chance performance (~50%) when force-rate cue is "inversely" applied in passive touch [26].

Third, in the "passive direct force-rate" task, where the softer stimulus was indented "directly" at a lower force-rate (0.5 N/s) than the harder stimulus (2 N/s), participants could differentiate the illusion case spheres near a 75% threshold (76.7% ± 5.4). This percentage of correct responses (with all participants aggregated) was significantly higher compared to the "passive inverse force-rate" task ($U = 0.5$, $p < 0.0001$, $d = 3.72$) and the "passive same force-rate" task ($U = 0.0$, $p < 0.0001$, $d = 4.33$). The values of participants' sensitivity were also improved for all stimulus pairs (detailed in S4 Table). These results empirically validate that, when force-rate cues are "directly" applied during the contact, cues besides those cutaneous become available in discriminating the illusion case spheres. It further indicates that the controlled force-rate cues may elicit alternate perceptible inputs and are likely perceived akin to proprioception, a point which will be detailed in the Discussion.

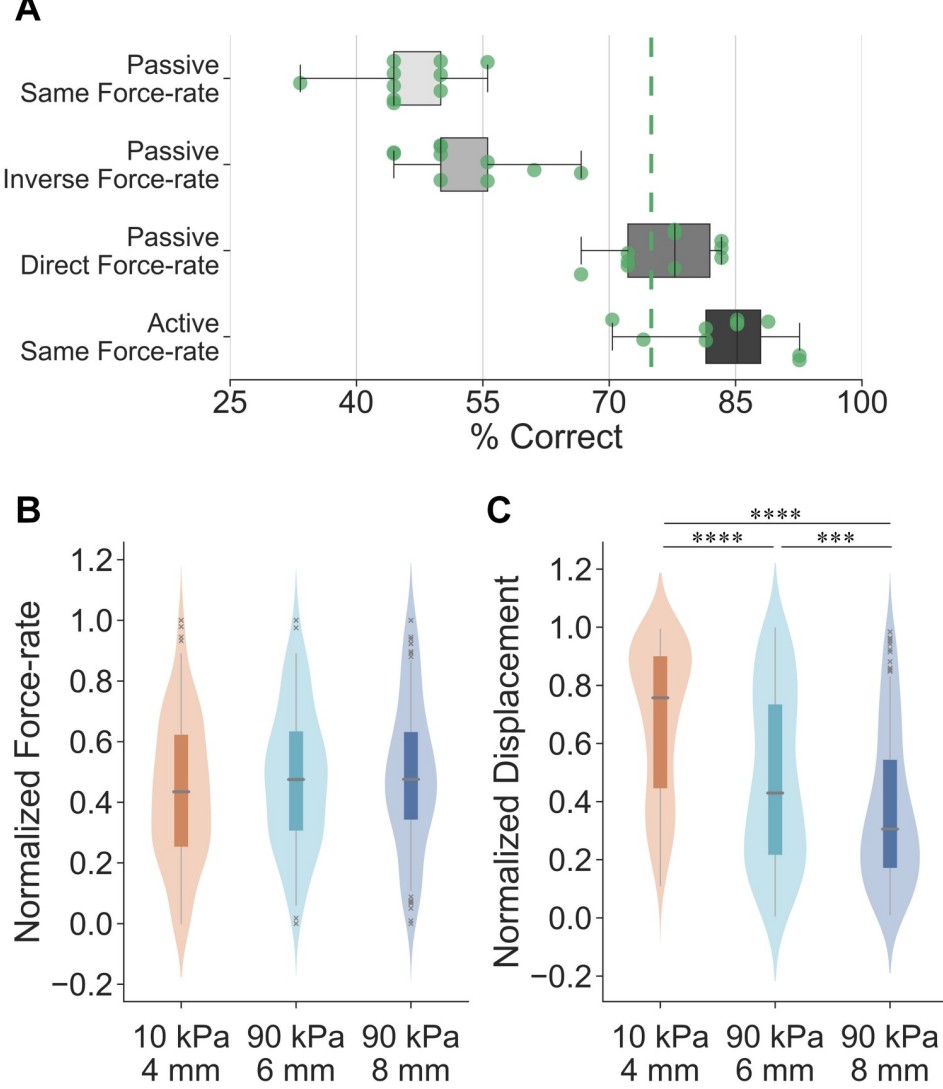

**Fig 6. Results of experiment 3: psychophysical evaluations and exploratory strategies.** (A) Psychophysical evaluations of illusion case spheres under different experimental conditions with all participants aggregated. The detection threshold is set as 75% for the same-different procedure. Points denote individual results. (B) Non-distinct force-rate cues are behaviorally applied for each illusion case sphere in active exploration of compliances. A miniature boxplot is set in the interior of the kernel density estimation of the underlying distribution. (C) Significantly higher fingertip displacement is applied for the small-compliant sphere, as opposed to the harder spheres. ***p < 0.001, ****p < 0.0001.

Fourth, to validate the hypothesis that the proprioceptive cue of active finger displacement may help to discriminate the illusion case stimuli, psychophysical evaluations were conducted in active touch. Participants ($n = 10$) were instructed to discriminate the illusion case spheres under fully active, behavioral sensorimotor control. Non-distinct force-rate cues were applied in exploring the illusion case spheres (Fig 6B), therefore, this experimental condition was denoted as "active same force-rate". As illustrated in Fig 6A, the spheres were readily discriminable with a percentage of correct responses of 83.7% ± 6.9 (detailed in S3 Table) and a mean sensitivity of 3.53 (detailed in S4 Table). This presents significantly better discrimination performance (with all participants aggregated) compared to the "passive direct force-rate" task

($U = 21.0$, $p < 0.05$, $d = 1.08$). Altogether, the proprioceptive cues elicited by active, volitional control of finger movements, help in discriminating the stimuli amidst indiscriminable cutaneous contact areas.

Furthermore, in active touch, participants volitionally move their fingers to generate consistent force trajectories between stimuli (Fig 6B) and thereby utilize the resultant differences in the fingertip displacements between the illusion case stimuli to discriminate them (Fig 6C). Specifically, given the same terminal indentation force level (2 N) and non-distinct force-rate cues (no significant difference detected) among illusion cases, significantly higher displacement was applied for the softer spheres (10 kPa-4 mm vs. 90 kPa-6 mm: $U = 8786.0$, $p < 0.0001$, $d = 0.74$; 10 kPa-4 mm vs. 90 kPa-8 mm: $U = 5737.5$, $p < 0.0001$, $d = 1.18$). This finding aligns with the finite element simulation where the 10 kPa-4 mm sphere exhibited higher fingertip displacement under the same force load (Fig 2D). In summary, when cutaneous cues as well as force-related movement cues are controlled, elicited differences in fingertip displacements help discriminate the illusion case spheres.

## Discussion

This study investigates an illusion phenomenon in exploring soft objects, specifically the situation in which small-compliant and large-stiff spheres are indiscriminable. These two physical attributes are common to everyday tasks; for example, in judging the ripeness of fruit. Through a combination of solid mechanics modeling, biomechanical contact measurement, and psychophysical evaluation, we show that small-compliant and large-stiff spheres afford nearly identical cutaneous contact, and thus, are indiscriminable in passive touch where only cutaneous cues are available. However, this phenomenon vanishes in active touch, when proprioceptive cues augment indiscriminable cutaneous contact cues. Furthermore, the results indicate that in the exploration of compliant objects, force-controlled movements are more efficient and optimal for eliciting the cutaneous and proprioceptive cues that underlie our judgments of compliance.

### A force-control movement strategy is optimal, efficient, and underlies softness perception

Amidst indiscriminable cutaneous contact cues, participants behaviorally control the exploratory forces they apply to soft objects. Specifically, the terminal indentation force, as well as the rate change of touch force was behaviorally controlled to be non-distinct among the illusion case spheres (Fig 6B). Indeed, participants actively move their fingers to apply consistent force trajectories and thereby evoke significant differences in fingertip displacement cues for softness discrimination. These fingertip displacements are proprioceptive by nature and critical to the discrimination of the illusion case stimuli (Fig 6C). Indeed, this exploratory strategy is important from a number of other perspectives. First, a force-modulation strategy is essential to compensate for the natural remodeling of the skin over time, which leads to changes in its thickness and elasticity [27]. Such changes in the skin's mechanics could generate large variance in neural firing patterns, and thereby perception. However, the skin can reliably convey information about indentation magnitude, rate, and spatial geometry when touch interactions are controlled by surface pressure. Since force directly converts to pressure on the skin upon contact, a force-modulation strategy echoes theories of active, behavioral control when exploring soft objects in daily tasks [27,29]. Second, at the behavioral level, we prioritize exploratory force to optimize our perception of object compliances in relevant contexts [29,36,40,41]. Indeed, the availability of force-related cues improves discriminability by reducing the necessary deformation of the skin [26]. Similarly, for the exploratory procedure of pinch grasp, we

control the grip force within a safety margin, informed by skin mechanoreceptors, to prevent slipping or applying exceedingly high pressure [42,43].

## Change of cutaneous contact as a cue to proprioception

As just discussed, the force-control movement strategy is efficient and optimal in evoking differentiable cues, in active touch. In passive touch, we observe that participants can discriminate the illusion case stimuli, particularly in the "passive direct force-rate" case, with a percentage of correct responses of about 77% (Fig 6A). While lower than the discrimination result for active touch, this represents a significant improvement over the "passive same force-rate" case, which yields chance performance.

We hypothesize that the modulation of force under "passive direct/inverse force-rate" condition–where the softer stimulus was indented "directly/inversely" at a lower/higher force-rate than the harder stimulus–provides an alternate perceptible input during the dynamic contact phase, also tied to finger proprioception. In particular, in alignment with prior findings [5,26], we show that the rate of change of force is linearly correlated with the rate of change of gross contact area (S6 Fig). While we cannot directly measure the rate of change of contact area, due to the limitations of the ink-based method only being able to measure terminal contact area, one could easily extrapolate this correlation to the dynamic contact phase by discretizing the terminal contact area/force into the instantaneous contact area/force. Using the 3D imaging technique, we indeed demonstrated that force-rate cue can proportionally elicit the instantaneous change of contact area [44]. Such cues might therefore induce the illusion of fingertip displacement amidst dynamic contact [37,45]. In particular, Moscatelli, *et al.* demonstrated that skin deformation of this kind naturally induces a sensation of relative finger displacement in the stationary hand [20,46]. Similarly, stretching the skin at the proximal interphalangeal joint can induce illusions of self-motion in anesthetized fingers [45]. Moreover, microscopic oscillatory stimulation at the skin surface also can elicit illusory finger displacements when pressing on a stiff surface [47]. Therefore, when passively exploring the illusion case spheres under the modulation of force-rate, the improved discriminability is likely derived from the proprioceptive sensation elicited by the change of contact area, which is originally induced by the force-rate cue.

Indeed, across a range of touch interactions broader than just softness, we find that cutaneous and proprioceptive cues are integrated to achieve high levels of performance [4,32]. In tasks involving reaching movements, cutaneous cues could systematically bias motion estimates, indicating that multisensory cues are optimally integrated for our motor control [4]. In general, multimodal interactions between these two signals are found to be mediated by distinct neural mechanism in primary somatosensory cortex [32]. These findings come in general agreement with prior studies reporting that both cutaneous and proprioceptive cues are needed in discriminating compliance. In particular, when finger movements are eliminated, our ability to discriminate pairs of spring cells decrease [28]. Likewise, when pinching an elastic substrate in-between two rigid plates, relatively lower discriminability of compliance is obtained when relying upon proprioception alone as compared to cutaneous cues alone [25].

## A perceptual illusion inspired by everyday tasks

The stimulus attributes of elasticity and curvature can be found in everyday, ecologically relevant tasks, e.g., judging the ripeness of a fruit for edibility. In some prior studies, however, stimuli have been highly engineered and delivered by sophisticated devices [10]. Such stimuli may not afford the same perceptual acuity as ecologically accurate soft objects [7]. Moreover, stimulus compliance at times has been parameterized by its stiffness rather than its modulus

[25,28,35], which can be confounding for naturalistic objects of identical stiffness but differing in geometry [35]. Herein, we address these issues by building spherical stimuli with covaried radii and elasticity which recapitulate important properties of ecologically compliant materials and mimic the contact profile of the skin surface's contacting elastic objects [9,10,22]. As it is difficult to measure the material properties of fruit, which can breakdown rapidly between sessions, our group has begun to consider the perceptual commonality between silicone-elastomer materials as reasonable stand-ins for ecological fruits [5]. Similar to the work with engineered substrates herein, we have found that the exploratory strategy of behaviorally controlling force aligns with how we judge the ripeness of fruit. In particular, we volitionally pinch soft fruit, by controlling grip force, to help differentiate their ripeness [5].

## Computational modeling formulates psychophysical studies

Instead of evaluating empirically with human-subjects a large number of stimulus combinations of elasticity and curvature, we computationally identified combinations with indistinct cutaneous contact. Indeed, a "computation first" effort as such demonstrates an alternative paradigm to bridge theoretical and empirical studies, make specific predictions and test particular hypotheses. Specifically, to better understand the encoding mechanism underlying the identified tactile illusion, cutaneous and proprioceptive cues need to be dissociated. As this is empirically demanding, we employed two interaction modes (passive and active touch, Fig 4) in the computational simulation. The potential cues and interaction modes that modulate the illusion are then validated in psychophysical experiments with human-subjects.

Finally and relatedly, far fewer illusions have been discussed in the tactile modality than for vision and audition [9,10]. This partially reflects the fact that tactile illusions are not as easily accessible [9]. Indeed, sophisticated efforts are usually required to create appropriate conditions to conceive the illusion, which is a significant electromechanical challenge to achieve empirically [10]. The "computation first" approach demonstrated herein may help in identifying potential illusions in a more efficient manner.

## Materials and methods

### Ethics statement

The human-subjects experiments were approved by the Institutional Review Board for the Social and Behavioral Sciences at the University of Virginia. Written informed consent was obtained from all participants.

### Geometry of the fingertip model

Two simplified 2D finite element models were derived from the geometry of a 3D model of the human distal phalanx bone [35]. The plane strain model of a cross-sectional slice from proximal first digit to distal tip was built for contact across the finger width (S4 Fig). Meanwhile, the axisymmetric model revolving around the centerline of the finger pad was built for contact normal to the surface (S4 Fig). Details of the model's structure and mesh are further explained in S1 Text.

### Material properties of the fingertip model

Hyperelastic material properties were used of the Neo-Hookean form of the strain energy function. The strain energy $\Psi$ was derived as:

$$\Psi = C_{10}(\bar{I}_1 - 3) + \frac{1}{D_1}(J - 1)^2 \tag{1}$$

where $C_{10}$, $D_1$ were material constants [35], $\bar{I}_1$ was the modified first strain invariant, and $J$ was the volume ratio known as the Jacobian matrix. The initial shear modulus $G$ was predefined and the initial bulk modulus was as $K = G/10^5$. The relationship between modulus and material constants were defined as $G = 2C_{10}$ and $K = 2/D_1$ accordingly.

The material elasticity was defined by its initial shear modulus $G$ which fully justified the material. Note that the material is in fact non-linearly hyperelastic. The Neo-Hookean model was applied to simplify the fitting procedure and derive a more robust calibration only based on the modulus $G$. Furthermore, instead of a linear Young's modulus, the hyperelastic form was considered for the soft objects which deform in a finite-strain region.

Finally, material calibration was conducted in two steps. First, the ratios of material elasticity between each layer were fitted to match the observed surface deflection to different displacements [48]. Second, the fitted ratios were scaled to fit the observed force-displacement relationships [49]. The detailed fitting procedures and final results are explained in S1 Text and S1 Table and S5 Fig.

### Stimulus tip model

Three values of radii (4, 6, and 8 mm) and elasticity (10, 50, and 90 kPa) were selected and the stimulus tips were modeled as hemispherical with the surface of central section attached to a rigid plate. The Poisson's ratio to the plate was set to 0.475 to mimic the nearly incompressible behavior of rubber. For the purpose of suppressing stress concentrations near nodes, triangular elements with 0.25 mm edge length were used in the region contacting the finger surface. Larger elements of up to 1.0 mm were used in non-contact region to lower the computational cost.

### Numerical simulations

Nine stimulus tips (3 radii by 3 elasticity) were built based on the 2D axisymmetric model and contact mechanics were simulated in an attempt to approximate passive and active touch interactions. In passive touch (Fig 4A), compliant stimuli were indented into the fixed fingertip at loads of 0.25, 0.5, 1, and 2 N. The response variables were derived as cutaneous cues only, quantified by stress distributions at the epidermal-dermal interface (470 μm beneath the skin surface), calculated by averaging neighboring elements at each interface node. Note that there were in total 111 element nodes employed to cover the locations from 0 to 15.2 mm. Proprioceptive cues were decoupled since the reaction force was provided by the fixture instead of muscle activity. In active touch (Fig 4B), the fingertip was ramped into the fixed stimuli to the aforementioned loads. The response variables were derived from both the cutaneous and proprioceptive cues. Specifically, the proprioceptive cue was approximated by the force-displacement relation of the fingertip in the normal direction. This measure is tied to the change of muscle length as detected by muscle spindles, while force indicates the change of the muscle tension of Golgi tendons [4,20,37].

### Stimuli and experimental apparatus

Nine compliant stimuli (3 radii by 3 elasticity) were constructed from a room temperature curing silicone elastomer (BJB Enterprises, Tustin, CA; TC-5005 A/B/C). To achieve the desired modulus, based on prior calibrations [22], corresponding ratios of cross-linker were added and mixed. These formulations were then cast into 3-D printed molds of three radii (4, 6, and 8 mm) and cured to become stimulus tips.

As illustrated in Fig 4A, a customized motion stage (ILS-100 MVTP, Newport, Irvine, CA) was built to indent the stimulus into the stationary finger pad [26]. Normal contact force was

recorded with a load cell (22.2 N, 300 Hz, LCFD-5, Omega, Sunbury, OH) mounted onto the cantilever. The 3D printed housing fixture was equipped with a servo motor (Parallax standard servo, Rocklin, CA) and actuator arms, enabling a quick switch between different stimuli. Customized circuitry and software were developed to command the indentations. Physical measures were employed to eliminate any movement of the finger pad during the indentation. First, the participant's forearm was supported by a stationary armrest bolted onto the base of the motion stage. Velcro straps were further used to constrain the forearm if any slipperiness was detected. Second, a plastic semicircular fixture was installed to hold the index finger. The inner diameter was determined based on the dimensions of participants' distal phalanx to fasten the distal and proximal interphalangeal joints. Finally, the finger pad was held at approximately 30 degrees relative to the stimulus surface.

The experimental setup for active touch is shown in Fig 4B. Instrumented load cells (5 kg, 80Hz, TAL220B, HTC Sensor, China) were installed on a fine-adjust rotary table which can be rapidly rotated to present the designated stimulus. To measure the fingertip displacement, a laser triangulation displacement sensor (10 μm, 1.5kHz, optoNCDT ILD 1402–100, Micro-Epsilon, Raleigh, NC) was mounted and the laser beam was calibrated to aim at the center of the stimulus surface. The forearm, wrist, and palm base rested on a parallel beam with no external constrains.

### Measurement of contact area

The gross contact area between the stimulus surface and finger pad was measured by the ink-based method [22,26]. An overview of this method is shown in Fig 4 and summarized as follows. At the beginning of each measurement, washable ink (Craft Smart, Michaels Stores, Inc., Irving, TX) was fully applied onto the stimulus surface. After each contact, the participant was instructed to gently indent the finger pad onto a blank section of a sheet of white paper, to fully transfer the stamped ink. The remaining ink on the finger pad was then completely removed. This procedure was repeated until all measurements were completed for the participant. The sheet of paper was then marked with a 5.0 cm reference bar and digitized for analysis. A center-radius pair was selected by the analyst to identify a region enclosing the fingerprint. The desired color rendering was adjusted to outline the edges from the background. Next, a serial search was conducted to find these bounding edges and the reference bar was also identified to scale the pixels. The final area was calculated using Gauss's formula in squared centimeters.

### Measurement of force and displacement

The gross contact readings from the force and laser sensor were smoothed to remove electrical artifacts by a moving filter with a window of 100 neighboring readings. The ramp segments of the force curves were then extracted based on first-order derivatives [5]. A linear regression was applied to the segments and the derived slope was noted as the force-rate. On the other hand, the fingertip displacement was calculated as the absolute difference between the initiation and conclusion of each movement.

### Participants

The human-subjects experiments were approved by the Institutional Review Board at the University of Virginia. Ten naïve participants were recruited (5 females and 5 males, 27.5 ± 2.6 years of age) and provided written informed consent. No history of upper extremity pathology that might impact sensorimotor function was reported. All participants were right-handed and were assigned to complete both the biomechanical and psychophysical experiments. All experimental tasks were completed and no data were discarded.

## Experiment procedure

In Experiment 2, the biomechanical measurement experiments were conducted in both passive and active touch with four stimuli (illusion case: 10 kPa-4 mm, 90 kPa-6 mm, and 90 kPa-8 mm; distinct case: 10 kPa-8 mm). For passive touch, all four stimuli were each indented into the finger pad at three force levels (1, 2, and 3 N) respectively. Each stimulus was ramped into the finger pad for one second and retracted away for one second. The ink-based procedure was applied for each indentation. There were three indentations for each stimulus at each indentation level per participant. All indentations were separated by a 20-second break. For active touch, the four stimuli were palpated by the index finger at three force levels which were behaviorally controlled. In particular, participants were instructed to actively press into the designated stimulus and a sound alarm was triggered to end the current exploration when their force reached the desired level. The ink-based procedure was used for each exploration. There were three explorations for each stimulus at each force level per participant. All explorations were separated by a 20-second break.

In Experiment 3, psychophysical discrimination experiments were conducted for both passive and active touch with the three illusion case stimuli. Following the rule of ordered sampling with replacement, nine stimulus pairs were drawn from the three illusion case spheres and were prepared for psychophysical evaluation. The stimulus ordering within each pair was determined by the sampling results (see S2 Table for detailed assignments). Participants were blindfolded to eliminate any visual information about the stimulus compliance or the movements of the indenter and the finger pad. No feedback on their performance was provided during the experiment. Using the same-different procedure, after exploring each pair (one touch per stimulus), participants were instructed to report whether the compliances of the two were the same or different. Note that the same-different procedure was applied herein because the observer can use whatever cues are available and does not have to articulate the ways in which the compliances actually differ [50,51]. This fits well with the experimental scope where the roles of perceptual cues are under investigation.

For passive touch, each trial consisted of discriminating one stimulus pair. Following the sampling order, spheres from the same pair were ramped into the fixed finger pad successively (Fig 4A). The indentation interval was controlled as 2-seconds to obtain consistent temporal effects on perception [52]. All discrimination trials were separated by a 15-second break. The terminal force level was set to 2 N as this aligned with Experiments 1 and 2. As illustrated in Fig 6, three experimental tasks were performed in passive touch. In the "passive same force-rate" task, all stimuli were indented at 1 N/s to 2 N. In the "passive inverse force-rate" task, higher force-rate was applied for the soft stimulus while the lower force-rate was applied for the hard stimulus. The 10 kPa-4 mm sphere was indented at 2 N/s to 2 N. The 90 kPa-6 mm and 90 kPa-8 mm sphere were indented at 0.5 N/s to 2 N. In the "passive direct force-rate" task, force-rate was applied in a direct positive relation with the stimulus modulus. The 10 kPa-4 mm sphere was indented at 0.5 N/s to 2 N and the two 90 kPa spheres were indented at 2 N/s to 2 N. For each experimental task, each of the nine stimulus pairs was presented twice. Adapted from prior studies [51,52], the test order of discrimination trials was randomized to balance the carry-over effects in response bias [53].

For active touch, the experiments were conducted under participants' fully active, behavioral control (Fig 4B). Within each discrimination trial, a participant was instructed to explore compliance by palpating each of two spheres successively with a terminal touch force of 2 N. When their force reached 2 N, a sound alarm was triggered to end that exploration. The interval between two explorations was set to 2-seconds as previously noted. Force and fingertip displacement were recorded simultaneously. Each stimulus pair was presented three times in a

randomized order to balance the carry-over effects in sequential responses. There was a 15-seconds break between trials. Note that trials under the same experimental task were grouped together and conducted within one block. Test order of the four experimental tasks (blocks) were randomized for each participant.

## Data analysis

As illustrated in Figs 5 and 6, the experimental results for all participants were aggregated for analysis. A normalization procedure was required for data aggregation since participants exhibited distinct sensorimotor capabilities, range of finger movements, and dimensions of the finger pad [5,26]. In particular, for each experimental task, all recordings of each tactile cue were normalized to the range of (0, 1) by sigmoidal membership function [5,30]. The center of the transition area was set as the mean value of the data normalized, and the logistic growth rate of the curve was set to 1. After this transition was completed for each participant, all results were then aggregated together for statistical analysis. The Mann–Whitney U test ($\alpha = 0.05$, two-sided test) was applied to compare the samples and the Cohen's $d$ (the absolute value) was calculated for statistically significant results to evaluate the effect size. The confidence interval was derived by bootstrapping the estimated data with 1000 iterations.

## Supporting information

**S1 Fig. Simulated spatial distributions of cutaneous cues.** (A) Spatial distributions of stress at contact locations for all nine spherical stimuli. (B) Spatial distributions of SED at the same contact locations for all spheres varying in radii and elasticity.
(TIF)

**S2 Fig. Cues of the surface deflection and finger displacement.** (A) Simulated surface deflection of nodes at the surface of the finger pad model for all the nine spheres. (B) Force-displacement relationships of the fingertip simulated for elasticity-radius combinations.
(TIF)

**S3 Fig. Average skin mechanics responses from the computational model over the same contact region for cutaneous cues.** For the intermediate force loads, average responses were quantified over the same contact region for tactile cues of (A) stress, (B) SED, and (C) surface deflection. The average stress/strain distributions overlap for the illusion case spheres, while similar average deflection cues were derived from all nine stimuli.
(TIF)

**S4 Fig. Geometry of the finger and stimulus tip model.** (A) The compliant stimulus is implemented as hemispheres contacting the skin surface of the finger pad. (B) Plane-strain model to fit the surface deflection. (C) Axisymmetric model to fit force-displacement relation and perform simulations. Adapted from [35] with permission.
(TIF)

**S5 Fig. Results of the material properties fitting.** (A) Relative ratios between skin layers are optimized to fit the surface deflection simulated by the model. The optimal point is selected by averaging all points with a $R^2 \geq 0.8$. (B) Force-displacement fits between model simulations and experimental measurements. Adapted from [35] with permission.
(TIF)

**S6 Fig. Perceptual cues measured in human-subjects experiments.** Gross contact areas measured in (A) passive and (B) active touch from one representative participant. Linear regression procedures are applied to visualize the correlation between touch force and contact area.

Translucent bands denote 95% confidence intervals for regression estimations. (C) Similar force-rates are volitionally controlled and applied in active exploration of illusion case spheres. (D) Distinct fingertip displacements are applied in discriminating the illusion case spheres.
(TIF)

**S7 Fig. Cutaneous and proprioceptive responses simulated during dynamic contact.** Stress distributions at contact locations for the three illusion case spheres: (A) 10 kPa-4 mm, (B) 90 kPa-6 mm, and (C) 90 kPa-8 mm. (D) Proprioceptive cues of finger displacement are simulated for all discretized force load during the ramp phase.
(TIF)

**S8 Fig. The rate of change in cutaneous responses during dynamic contact.** Derived from S3 Fig, the rate of change of averaged (A) stress, (B) SED, and (C) surface deflection are calculated for the contact ramp phase. Note that within the simulation procedure, time points are linearly coupled with force loads, i.e., 0.5 N is applied at 0.25 sec and 1.5 N is applied at 0.75 sec, etc.
(TIF)

**S1 Table. Material properties derived from the fitting.** The final shear moduli for the skin layers are taken as the average of all subjects' results. Adapted from [35] with permission.
(XLSX)

**S2 Table. Stimulus pairs drawn from the three illusion case spheres.** Nine stimulus pairs are drawn from the three illusion case spheres for psychophysical experiments. Stimulus ordering within each pair is determined following the rule of ordered sampling with replacement.
(XLSX)

**S3 Table. Results of psychophysical evaluations of all nine stimulus pairs.** Percent correct responses for each stimulus pair under different experimental conditions with all participants aggregated. Note that the ordering within each pair was consistent with S2 Table.
(XLSX)

**S4 Table. Signal detectability of the three illusion case spheres.** The sensitivity measure, *d'*, is derived from the hit and false-alarm rates, providing a bias-free measure of detectability. Under the assumption of differencing rule, *d'* values for each condition are determined from Table A 5.4 in [39].
(XLSX)

**S1 Text. Supporting text.** The text includes five sections: perceptual cues predicted in the computational modeling, geometry of the fingertip model, fitting hyperelastic material properties, perceptual cues measured from one representative participant, and perceptual cues predicted during the dynamic contact.
(DOCX)

## Acknowledgments

We would like to thank all the participants of the human-subjects experiments, as well as members of the Gerling Touch Lab for fruitful discussions and feedback. The content is solely the responsibility of the authors and does not necessarily represent the official views of the National Institutes of Health or National Science Foundation.

## Author Contributions

**Conceptualization:** Chang Xu, Yuxiang Wang, Gregory J. Gerling.

**Data curation:** Chang Xu, Yuxiang Wang, Gregory J. Gerling.

**Formal analysis:** Chang Xu, Yuxiang Wang, Gregory J. Gerling.

**Funding acquisition:** Gregory J. Gerling.

**Investigation:** Chang Xu, Yuxiang Wang, Gregory J. Gerling.

**Methodology:** Chang Xu, Yuxiang Wang, Gregory J. Gerling.

**Project administration:** Gregory J. Gerling.

**Resources:** Gregory J. Gerling.

**Software:** Chang Xu, Yuxiang Wang, Gregory J. Gerling.

**Supervision:** Gregory J. Gerling.

**Validation:** Chang Xu, Yuxiang Wang, Gregory J. Gerling.

**Visualization:** Chang Xu, Yuxiang Wang, Gregory J. Gerling.

**Writing – original draft:** Chang Xu, Yuxiang Wang, Gregory J. Gerling.

**Writing – review & editing:** Chang Xu, Yuxiang Wang, Gregory J. Gerling.

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
