## [Decision Letter · Decision Letter 0]

4 Nov 2020

Dear Prof. Gerling,

Thank you very much for submitting your manuscript "An elasticity-curvature illusion decouples cutaneous and proprioceptive cues in active exploration of soft objects" for consideration at PLOS Computational Biology.

As with all papers reviewed by the journal, your manuscript was reviewed by members of the editorial board and by several independent reviewers. The reviewers appreciated the novelty of the work, but raised some concerns about the psychophysical experiments and framing of parts of the paper. In light of the reviews (below this email), we would like to invite the resubmission of a significantly-revised version that takes into account the reviewers' comments.

We cannot make any decision about publication until we have seen the revised manuscript and your response to the reviewers' comments. Your revised manuscript is also likely to be sent to reviewers for further evaluation.

Sincerely,

Adrian M Haith

Associate Editor

PLOS Computational Biology

Samuel Gershman

Deputy Editor

PLOS Computational Biology

Reviewer's Responses to Questions

**Comments to the Authors:**

Reviewer #1: The authors describe a set of experiments in which they have fabricated a set of stimuli whose elasticity and curvature vary so as to explore the contact mechanics between the objects and the finger and the perceived softness of the objects. Computational models of the contact mechanics between the fingertip and spheres of varying radii and elasticity were initially developed to capture the stress distributions and finger surface deflection during contact. Based on these models, particular stimuli were selected for the biomechanical and psychophysical experiments. The results are of interest in that they demonstrate the critical role of proprioceptive cues in perceiving the compliance of manually explored objects and of the rate of change in contact area when objects are sensed under passive touch conditions.

One major issue with the manuscript is how the models developed and experiments conducted are framed. The authors refer to their development of an illusion which is not an appropriate approach to understanding human perception. A perceptual illusion refers to a misperception or a discrepancy between a physical stimulus and its corresponding percept. It is not that an illusion is developed a priori, more that a set of stimuli are fabricated (by varying elasticity and curvature in this case) that have particular properties based on the FEMs of the fingerpad and object that could result in errors in perception. Once these “errors” have been demonstrated, that is that there is an illusion, the focus is on understanding the mechanisms that account for the perceptual errors. By analogy to the size-weight illusion that the authors refer to, the stimuli designed for experiments studying this illusion are not by definition illusory, volume and mass are covaried to create a stimulus set. Similarly, the term “illusory cutaneous contact” does not accurately depict the mechanics of finger contact which are not illusory.

The authors propose that because the spheres are indiscriminable when explored passively they create an illusory experience and that when explored actively the illusion disappears. The “illusion” is not a property of the spheres but of how they are explored. With active touch there is no illusion and so it does not seem to make sense to refer to these spheres as illusory. Similarly with the passive direct-force rate presentation the spheres were now discriminable.

Detailed corrections:

p.3 line 36: cutaneous response – responses

p.3 line 57: With it – what does “it” refer to?

p.4 line 61: in analogy – by analogy

p.4 lines 62-63: the tactile/proprioceptive illusion described here should have more detail so that the reader can understand the effect – the speed of a moving stimulus changes depending on whether it is perceived via tactile motion or with arm movement.

p. 4 line 77: In what way does the tactile illusion “underlie our perception of softness” which implies that the illusion is fundamental to perception?

p. 5 line 82: If the illusion is observed only under passive touch conditions how does it “naturally decouple cutaneous cues from proprioceptive movements” given the latter do not occur with passive touch?

p. 5 lines 85-86: Is it really correct to state that “small-compliant and large-stiff spheres are perceived as identical?” The fact that participants cannot discriminate reliably between them does not mean they are perceived as identical. The results show that they cannot be discriminated.

p. 5 line 86: “This is done for single, bare-finger touch” – What is done?

p. 12 line 216: remove likewise

p. 13 line 220: “while the others were quite distinct from 1.48…” - do you mean averaged 1.48 cm2 ?

p. 13 line 226: cutaneous contact is indiscriminable – Experiment 2 focused on biomechanical measurements and so the term indiscriminable is not appropriate in this context. The results indicated that there was not a significant difference in the contact areas for the illusion-case stimuli.

p. 13 line 233: but their contact areas found non-differentiable – clumsy sentence, rewrite

p. 14 line 243: Can an illusion become discriminable? It is the stimuli being explored under certain conditions that makes them discriminable. (also line 257)

p. 15 line 263: Why is cutaneous contact illusory? (also line 284)

p. 17 line 298: What does “overly grasped” mean?

p. 17 line 314: displacement to the – displacement in the

p.18 line 338: What is the evidence that this illusion “is naturalistic and commonplace in (not as) our daily lives?”

p. 19 line 364: phalange should be phalanx (singular form of phalanges)

p. 24 line 468-469: It is not clear how randomization of stimuli has an effect on fatigue or inattention? There are various biases in participants’ responses that can occur with fixed sequences of stimulus presentation which is why they are typically randomized in perceptual experiments.

p. 25 lines 481-483: To be consistent with how the term is typically used in psychophysical experiments, the term “trial” should refer to the presentation of one stimulus pair which were presented successively separated by 2 s. From each trial one response was provided by the participant. The time interval between trials was 15 s.

p. 25 line 488: remove some of the instances of “each”

Reviewer #2: The ms present a computational model of stimuli and finger pad from which a perceptual illusion is derived. The existence of the illusion is tested in a psychophysical experiment.

I very much like the approach to derive a potential perceptual illusion from the model, which sounds highly convincing to me. Also the first biomechanical test (Exp. 2) is nice and state-of-the art. In my view, however, the psychophysical experiment is confound because it does not carefully separate perceptual sensitivity from response bias. In addition the ms is not easy to read in places, in particular because important information is given too late. Also, implications of the results for decoupling cutaneous and proprioceptive cues are not well discussed. With new experimental data, however, the paper may definitely be worth to be published in a PLOS Comp BIOL.

Confound in Exp 3:

A same/different task was used here, and the percentage of difference response was taken as a measure of perceptual performance. In this task, however, the percent “different” do not only depend on perceptual precision, but also on response tendencies (that is, the tendency to respond same or different in unclear cases). Typically, a control condition with two equal stimuli presented under equal conditions is included in order to dissociate perceptual precision from response tendency (cf. e.g. Green, D.M., Swets J.A. 1966, Signal Detection Theory and Psychophysics. New York: Wiley). In the present case, one control condition would have been required for each of the four conditions (passive with different force rates, active). However, as these control conditions were not included results may alternatively be explained by different response tendencies, e.g. by speculating that the less regular stimulations patterns in active vs.passive touch increase the percentage of “different” responses. I strongly recommend to redo experiment 3 including control conditions (and calculate perceptual precision separate from response tendency as described, e.g. in Green and Sweets, 1966) or, maybe better, to redo it using a response-bias free discrimination task (Which stimulus was softer?”).

Comprehensibility

Abstract: Is not easy to read, might focus more on what has actually done.

P3, 58: Please extend a bit on the conditions of the curvature illusion.

P4, 62-63: The Doppler effect is a physical phenomenon, not a psychological one, and hence does not fit here. Please extend also a bit on the phenomenon reported in [15]

P4, 64-65: How do these phenomena shed light upon perceptuo-sensorimotor dependencies? Please extend. Please extend also a bit on applications.

P4, last para: The illusion needs to be better explained here.

Fig. 2: How precisely is the location (x-axis) defined?

Fig. 5A, B: Please indicate to what a single data point refers to (given there are several data points per condition)

P13, 219, 223: Which data entered the statistical tests and which and how many tests were used? Please indicate here.

P13, 232: What exactly was the task/instruction of participants? Why is the result called a “detection rate” rather than a percentage of correct discriminations?

P14, 246: What is exactly meant by a “higher force-rate” (over time)? And how much higher was it? Overall, the rationale of the force-rate manipulation is unclear at this point of the ms and should be better explained. It is also unclear how the active condition (Fig. 6A) can be a “same-force rate” condition

P14, 248: Which data entered the statistical test and which test was used? Please indicate here.

Exp. 3: How was it exactly achieved that the finger was not differently displaced in the passive conditions, and how successful were these measures? Please report in the main ms.

P21, 407, and P23, 444: How many stimuli were made out of silicone? Only four or the nine from the simulations? Which were used?

P24, 465-477: How was the order of trials from different (passive) conditions? Were they blocked or random? How was the order of stimuli in each trial?

Further points

Fig. 2, Fig. 3: it is very nice to see similarity and dissimilarity of stress distributions and other measures here. I wonder whether in addition an aggregate number of (dis-)similarities was available that should then be reported for all 9 stimuli.

P16, Discussion: please explain in greater detail how you conclude from your data on the strategy of volitional force control.

P17, 313-314, “… skin deformation of this kind naturally includes a sensation of relative finger displacement … “: This effect should occur for both the passive direct force-rate, and the passive indented force-rate condition, and hence cannot separately explain the behaviour in the passive direct force-rate condition, as done here.

P24, 456-457, “nine stimulus pairs were drawn from the three illusion cases”: Please explain in greater detail. Make clear what the three cases are and how nine pairs can be drawn from three cases.

P24, 465-477, from these lines and table S2 in the supplement I concluded that the three cases were 10 kPa-4mm with 90kPA 8mm, 10 kPa-4mm with 90kPA 6mm, and 90 kPa-6mm with 90kPA 8mm. In the latter pair, compliances of the two stimuli were the same. Was the response recoded in that case? That is was a “same”-response for that pair considered a “correct” response in fig. 6 A? In table S2 it appears that this was (incorrectly) not the case, given that in the passive same force ratio condition the percentage for the latter pair was as low as for the other two pairs. Please clarify.

Reviewer #3: This study investigates the sensory cues involved in the perception of softness. Specifically, using computational modelling, a novel illusion is identified, whereby objects that differ in elasticity will feel the same when indented passively into the skin, but will feel different when touched actively using comparable forces. The illusion works by employing objects of different sizes such that their tactile imprint onto the skin is identical, while additional proprioceptive input in the active condition will disambiguate between the objects.

The paper is well-written and easy to follow. The study itself is cleverly designed and it is especially great to see the interplay between computational modelling and psychophysical measurements used to establish a novel illusion. My comments mainly focus on clarity and context.

1) The study focuses exclusively on the static part once the force has reached its plateau. However, many tactile afferents respond exclusively to the dynamic period and the overall population firing rate might well be higher in the initial contact phase than in the plateau phase. Would the skin mechanics model predict that the illusion should also be apparent during this phase? Or would contact differ instead, but for some reason this difference might not translate into a perceptual effect? Further detail is needed here and this problem should be discussed and ideally addressed with further analysis.

2) Figure 1: It would help here to explicitly show the dermis/epidermis border in the illustration to indicate where stress mostly impacts neural responses.

3) Figures 1, 2, 3: It is not clear why a 90 kPA / 6mm stimulus is included in the four examples, rather than 90 kPA / 4 mm, which would mirror the other stimuli (all possible combinations of 10/90 kPa and 4/8mm). This should be fixed / explained.

4) In the psychophysical results, performance is “significantly improved” in the passive condition with the inverse force rate. However, are any of the passive results actually different from baseline (50%)? If anything, it appears to me that the results in the passive inverse condition are closer to baseline than in the other passive condition. Similar to point 1) above, different force rates would also lead to different tactile feedback during the initial dynamic part of the contact.

**Have all data underlying the figures and results presented in the manuscript been provided?**

Reviewer #1: Yes

Reviewer #2: None

Reviewer #3: **No: **Data appear to be available on Figshare, but link provided did not work for me.

PLOS authors have the option to publish the peer review history of their article (what does this mean?). If published, this will include your full peer review and any attached files.

Reviewer #1: No

Reviewer #2: No

Reviewer #3: No
---

## [Decision Letter · Decision Letter 1]

5 Feb 2021

Dear Prof. Gerling,

Thank you very much for submitting your manuscript "An elasticity-curvature illusion decouples cutaneous and proprioceptive cues in active exploration of soft objects" for consideration at PLOS Computational Biology. The reviewers were overall very satisfied with the revisions to the manuscript. However, Reviewer 1 noted several minor points that warrant attention before proceeding.

Sincerely,

Adrian M Haith

Associate Editor

PLOS Computational Biology

Samuel Gershman

Deputy Editor

PLOS Computational Biology

[LINK]

Reviewer's Responses to Questions

**Comments to the Authors:**

Reviewer #1: Line 19: constrained stationery – constrained to be stationery

Line 20: are made perceptible – are perceptible

Line 25: cutaneous contact – cutaneous contact cues

Line 46: comparing to prior experiences – comparing what?

Lines 57-58: could be perceived as being convex or concave curved – can be perceived as being convex or concave (remove “curved”)

Lines 61-63: Explain the task in more detail this is very cryptic

Lines 67-68: This should be referred to as the pseudo-haptic effect

Line 85: What are non-distinct cutaneous cues?

Lines 162-163: therefore naturally dissociate from proprioceptive movements – is the point that the cutaneous cues are similar and therefore non-informative whereas proprioceptive cues may yield useful information about contact?

Line 196: profile – profiles

Line 234: no significantly difference – no significant difference (also on lines 249, 311)

Line 269: Meanwhile… - this implies that while the first task was occurring something else was being implemented which was not the case – remove meanwhile (also line 391)

Line 274: yielded a chance performance – the d’ value indicates that performance is at chance is does not yield it

Line 282: discrimination correctness – “percentage of correct responses” seems a better term than discrimination correctness, average percentage of correctness and result of correctness (all in several places, lines 350-352)

Line 283 and elsewhere in the Results section: I assume the “averages” reported are means? If so, they should be referred to as means (or modes or medians).

Line 319: specifically where – specifically the situation in which

Line 325: cues akin to proprioception – why are the cues “akin”?

Line 332: “of touch force was volitionally controlled to be non-distinct among the illusion case”: using the word volitionally in this context does not seem appropriate – the touch force was controlled but it is not clear that participants were consciously aware that they were deliberately controlling this force which is what the concept of volition entails.

Line 345: improve – improves (the availability….improves)

Line 377: This comes in general agreement – what comes?

Line 446: in attempt -in an attempt

Line 471: Velcro straps were further used to constrain if any slipperiness was detected. – constrain what? The forearm?

Reviewer #2: I carefully checked through the authors changes. They really did a good job in responding to my concerns. From my side, there is nothing to quibble about. Hence, I recommend acceptance. Only a minor point: Maybe the author might want to recheck their Mann-Whitney U-statistics, because it quite often was 0.

Reviewer #3: All my concerns have been addressed. I would like to thank the authors for providing manuscript changes within the rebuttal, making it very easy to evaluate changes. I look forward to seeing the paper in print!

**Have all data underlying the figures and results presented in the manuscript been provided?**

Reviewer #1: Yes

Reviewer #2: Yes

Reviewer #3: None

PLOS authors have the option to publish the peer review history of their article (what does this mean?). If published, this will include your full peer review and any attached files.

Reviewer #1: No

Reviewer #2: No

Reviewer #3: No
---

## [Editor Report · Decision Letter 2]

3 Mar 2021

Dear Prof. Gerling,

We are pleased to inform you that your manuscript 'An elasticity-curvature illusion decouples cutaneous and proprioceptive cues in active exploration of soft objects' has been provisionally accepted for publication in PLOS Computational Biology.

Best regards,

Adrian M Haith

Associate Editor

PLOS Computational Biology

Samuel Gershman

Deputy Editor

PLOS Computational Biology

---

## [Editor Report · Acceptance letter]

17 Mar 2021

PCOMPBIOL-D-20-01703R2 

An elasticity-curvature illusion decouples cutaneous and proprioceptive cues in active exploration of soft objects

Dear Dr Gerling,

I am pleased to inform you that your manuscript has been formally accepted for publication in PLOS Computational Biology. Your manuscript is now with our production department and you will be notified of the publication date in due course.

With kind regards,

Alice Ellingham
